# Percutaneous Electrolysis in the Treatment of Lateral Epicondylalgia: A Single-Blind Randomized Controlled Trial

**DOI:** 10.3390/jcm9072068

**Published:** 2020-07-01

**Authors:** Manuel Rodríguez-Huguet, Jorge Góngora-Rodríguez, Rafael Lomas-Vega, Rocío Martín-Valero, Ángeles Díaz-Fernández, Esteban Obrero-Gaitán, Alfonso Javier Ibáñez-Vera, Daniel Rodríguez-Almagro

**Affiliations:** 1Department of Nursery and Physiotherapy, University of Cádiz, 11009 Cádiz, Spain; manuel.rodriguez@uca.es; 2Hospital de La Línea de la Concepción, 11300 Cádiz, Spain; 3Policlínica Santa María Clinic, 11008 Cádiz, Spain; jorgem.gongora@gmail.com; 4Department of Health Sciences, University of Jaén, Campus las Lagunillas, 23071 Jaén, Spain; rlomas@ujaen.es (R.L.-V.); andiaz@ujaen.es (Á.D.-F.); eobrero@ujaen.es (E.O.-G.); dralmagro4@gmail.com (D.R.-A.); 5Department of Physiotherapy, Faculty of Health Sciences, University of Málaga, 29071 Málaga, Spain; rovalemas@uma.es

**Keywords:** electrolysis, elbow tendinopathy, tennis elbow, lateral epicondylitis, lateral epicondylalgia, physical therapy

## Abstract

Few studies have considered the effects of percutaneous electrolysis (PE) in the treatment of lateral epicondylalgia (LE). For this reason, the objective of this study was to compare the effects of PE with an evidence-based approach—trigger point dry needling (TDN)—in patients with LE. A randomized controlled trial was conducted in which 32 participants with LE were randomly assigned to two treatment groups, the PE group (*n* = 16) and the TDN group (*n* = 16). Both groups received four therapy sessions and an eccentric exercise program to be performed daily. The numerical pain rating scale (NPRS), pressure pain thresholds (PPT), quality of life, and range of motion were measured before treatment, at the end of treatment, and at one- and three-month follow-ups. Significant between-group mean differences were found after treatment for NPRS (*p* < 0.001) and flexion movement (*p* = 0.006). At one-month follow-up, significant mean differences between groups were found for NPRS (*p* < 0.001), PPT (*p* = 0.021), and flexion (*p* = 0.036). At three-months follow-up, significant mean differences between groups were found for NPRS (*p* < 0.001), PPT (*p* = 0.004), and flexion (*p* = 0.003). This study provides evidence that PE could be more effective than TDN for short- and medium-term improvement of pain and PPTs in LE when added to an eccentric exercise program.

## 1. Introduction

Lateral epicondylalgia (LE) is defined as a musculoskeletal pathology with pain in the lateral area of the elbow, pain on extension of the wrist, and reduced grip strength [1]. It is most prevalent among individuals aged 35–50 years and affects approximately 1 to 3% of the general population [1,2,3]. Forty percent of the population may be affected by this condition at some point in their lives [3], and the risk is the same in both sexes [4]. This pathology is related to repetitive movements of the elbow, forearm, and wrist, and is usually derived from sports activity and activity in the workplace, with particular prominence in industrial professions [5].

Most of the time, this injury is considered as a tendinopathy because lateral epicondylalgia often presents with a degenerative process that affects the tendon structure, although motor system impairments and changes in pain perception have also been observed [6,7]. Changes can be seen at the histological level with disorganization in the arrangement of collagen fibers and local hypervascularization, proliferation of fibroblasts, and tissue granulation [8,9,10]. Specifically, in LE, there are changes in the tendon of the extensor carpi radialis brevis muscle [7,9,11], and usually, the pain is intensified by resisted muscular contraction of the extensor carpi radialis brevis [3,7]. On ultrasound examination, it is possible to observe thickening of the tendon, areas of hypogenicity, and neovascularization [4], as well as the existence of angiofibroblastic hyperplasia [7]. On manual assessment, pain increases on palpation, and range of motion is limited [3]. However, physical examination should not be limited to the elbow area and should include specific tests, such as the Thompson maneuver, Mill’s test, Chair test, Bowden test, and Cozen’s test [7].

The most commonly used treatment for LE is rest, non-steroidal anti-inflammatory drugs, splints, physical therapy, injection therapy, and surgery, although none of these are universally effective [1,7]. A corticosteroid injection is the most used invasive therapy due to its low cost and easy application; however, its positive effects in reducing pain and increasing functionality are only short term [1]. Within physical therapy, trigger point dry needling (TDN) [12] and eccentric exercises are used in tendinopathies to improve activation of mechanoreceptors as well as fibroblast stimulation and collagen synthesis [4,13]. Moreover, there are innovative minimally invasive techniques such as percutaneous electrolysis (PE). PE involves the application of a local percutaneous galvanic continuous current using an ultrasound-guided acupuncture needle that transmits the current directly to the degenerated tissue, producing a focused inflammatory response leading to its regeneration by proliferation of new collagen fibers at the injured tissue [13,14,15].

Currently, there are a few studies that support the effectiveness of percutaneous electrolysis in LE [6]. However, these trials do not include a control group and do not measure the range of motion of the elbow joint. Therefore, the present study is necessary to delve into its effects. The present study was proposed with the objective to evaluate the effects of PE on pain and range of movement compared to treatment with myofascial TDN on the epicondylar musculature in patients with LE. Treatment of LE with TDN has shown improvements in local pain and disability in elbow tendinous injuries [12], therefore, this minimally invasive treatment could be considered an adequate control for another minimally invasive technique, such as PE.

The hypotheses of the study were that treatment with percutaneous electrolysis added to an eccentric exercise protocol would improve range of motion and pain in patients with LE, achieving more considerable improvement compared to treatment with TDN on the myofascial trigger points of the epicondylar musculature added to the same protocol of eccentric exercises.

## 2. Material and Methods

### 2.1. Study Design

A single-blind randomized controlled clinical trial was designed. This study was conducted between July and September 2018. Before beginning the study, all participants signed an informed consent according to the Helsinki Declaration. Ethical approval for the study was obtained from the Ethics Committee of Research of Cádiz (Spain) (reference number 55/17). The trial was registered in the Clinical Trials Registry (reference number NCT03225404). The current study conforms to the CONSORT statement for reporting clinical trials [16].

### 2.2. Participants, Randomization, and Blinding 

Participants were recruited from the Santa María Clinic (Cádiz, Spain). A researcher, who was blinded to which treatment participants received, conducted an informative session about their rights to withdraw from the study and potential risks associated with the intervention. After this, the researcher collected the relevant data from those agreeing to participate. The inclusion criteria included: patients of both sexes, aged between 18 and 60 years, and diagnosed with LE with a poor evolution (pain reduction lower than 2 points on the Numerical Pain Rating Scale (NPRS) after one month of passive physiotherapy TENS, and stretching exercises) and pharmacological treatment. The exclusion criteria for this study were as follows: patients who were pregnant, those who had pacemakers, those who had undergone local surgery at the elbow, those who were treated with percutaneous electrolysis a month earlier, fibromyalgia patients, subjects with cervical radiculopathies, and patients with cancer or infectious processes. The participants were randomly allocated to each intervention group using a 1:1 allocation ratio. The randomized sequence for allocation was created by an independent researcher using a random allocation software program (Epidat 4.0) and was concealed in sequentially numbered envelopes.

### 2.3. Sample Size Calculation 

The sample size was obtained with the scientific software Epidat (Epidat: Epidemiological Analysis of Data, Version 3.1, January 2006, Consellería de Sanidade, Xunta de Galicia, Santiago de Compostela, Spain). The calculations were performed taking into account a detection of differences between groups of 1.1 units of NPRS, corresponding to the minimal clinically important difference found for this scale [17], with a standard deviation in post-treatment of 1.0 and 1.2 units of both groups [13], a 95% confidence level, and a statistical power of 80%. With these data, the required sample size was 16 subjects per group and a combined total of 32 subjects included in the study.

### 2.4. Outcome Measures

All measurements were conducted by a well-trained physician who was blinded as to which group the patient belonged to. The pain intensity of LE was measured with an NPRS considering 0 as no pain and 10 as the worst pain. This scale has demonstrated an intraclass correlation coefficient (ICC) of 0.61 to 0.95, a Standard Error of Measurement (SEM) of 0.48 to 1.02, and a Minimum Detectable Change (MDC) of 1.32 to 2.8 points for musculoskeletal pain disorders [18,19,20].

The pressure pain threshold (PPT) is defined as the minimum amount of pressure needed to cause pain at the point of activation. This outcome was evaluated with a pressure algometer (pain test FNP 100, Wagner Instruments^®^, Greenwich, USA) at the epicondyle according to the methodology described by Koçak et al. [21]. The doctor located the painful point at the lateral epicondyle, placed the tip of the algometer perpendicular to the skin, and applied pressure that gradually increased by 1 kg/cm^2^ per second. Patients were instructed to indicate if they felt local or referred pain and to stop at the point where the pressure became painful. The mean of three non-consecutive measurements, with a rest interval of 30 s, was chosen as the reference value. Pressure algometry is a valid and reliable tool for assessing pain and has demonstrated fair to excellent reliability with an ICC ranging from 0.78 to 0.93 [22].

Another variable measured was elbow joint motion. For this purpose, a digital inclinometer was used (Baseline^®^, India), a tool frequently used in previous studies to achieve the same goal [23]. This tool has shown good reliability and an MDC of 4° to 9° to be sure that change is not due to inter-trial variability or measurement error [23]. The inclinometer was placed on the lateral surface of the elbow, in line with the arm, to measure flexion and extension and at the third metacarpal bone to measure pronation and supination [24].

Quality of life was assessed with the SF-12 questionnaire, which is the short version of the SF-36. This twelve-question instrument includes two components: the physical (PC-SF12) and the mental (MC-SF12). This tool has demonstrated good reliability in the Spanish population (Cronbach’s alpha of 0.85 for PC-SF12 and 0.78 for MC-SF12) [25].

### 2.5. Interventions

The interventions in each group consisted of an invasive physiotherapy procedure followed by a common eccentric exercise protocol, whose objective was to modulate the load on the common tendon of epicondylar musculature insertion. The participant lay in a supine position with the affected elbow in a position of 90° flexion and pronation. The physiotherapist, equipped with sterile gloves, marked the location of the area to be treated with a demographic pencil, and the puncture area was disinfected with alcohol in both treatment groups. The duration of the intervention and follow-up was identical in both groups. 

#### 2.5.1. Experimental PE Group 

The application of the percutaneous electrolysis technique was performed with an EPTE^®^ percutaneous electrolysis device (Ionclinics & A. Deionic SL, Valencia, Spain) for 1.2 min at an intensity of 350 µA in the insertional tendon of the muscles of the epicondyle (Figure 1) using a 0.3 mm needle guided by ultrasound (Voluson 730 pro, General Electric^®^, Boston, MA, USA) and forming an angle of between 30° and 45° with the axis thereof (Figure 2). The treatment was performed once a week for four weeks [6].

#### 2.5.2. Control Trigger Point Dry Needling (TDN) Group

TDN was performed in the epicondylar musculature (supinator muscle and extensor carpi radialis longus muscle) in a relaxed perpendicular position with the forearm pronated, directing the needle toward the radius [26]. This procedure was applied once per week for four weeks.

#### 2.5.3. Eccentric Exercise Protocol for Both Groups

Patients in both groups were taught the eccentric exercises for the epicondylar musculature that were to be performed daily at home, from the first day until the last day of treatment. The eccentric exercise protocol consisted of three series of ten repetitions of eccentric work twice daily (morning and afternoon) with 1 kg weights. With the elbow in extension and the forearm in pronation, the movement was maximum flexion and wrist extension, restoring initial position passively with the other hand to wrist extension and come back to flexion with eccentric work [6,27,28,29]. To monitor compliance, participants had to fill out a daily registry with the time when they performed the exercises.

### 2.6. Statistical Analysis 

Data management and analyses were carried out using the SPSS statistical package, version 23.0 (SPSS Inc., Chicago, IL, USA). The level of statistical significance was established as *p*-value < 0.05. The data were reported using mean and standard deviation for continuous variables, and frequencies and percentages for categorical variables. The Kolmogorov–Smirnov test was used to determine the normality of the continuous variables. Levene’s test of equality of variances was used to determine the homoscedasticity of the variables.

To examine the baseline differences between groups in terms of the morphological characteristics and the study variables, the Student’s t-distribution was used for the variables that met the assumptions of continuity and homoscedasticity, and the nonparametric Mann–Whitney U-test was used for the variables that did not meet these assumptions.

Due to the lack of continuity and homoscedasticity of some of the study variables, the differences between groups were analyzed based on the change in scores between the initial measurements and the post-treatment measurements: the initial measurements and the measurements one month after treatment, and initial measurements and measurements three months after treatment. For this analysis, the analysis of variance (ANOVA) test was used for the variables that met the assumptions of continuity and homoscedasticity, and the nonparametric Mann–Whitney U-test was used for the variables that did not meet these assumptions. The eta-squared statistics (η2) were used to describe the effect size.

Based on the article by Mishra et al. [30], a 25% improvement in the patient’s pain perception, according to the NPRS scale, was defined as clinical success. However, the authors highlighted in the same publication that a 25% improvement might not be considered clinically significant after 24 weeks of treatment. Therefore, clinical success was proposed to be calculated as a 50% improvement on the NPRS scale for the three-month follow-up after treatment. The calculation was performed using NPRS score differences between baseline and after treatment of each participant.

## 3. Results

All participants completed the planned evaluations and treatments both at home and at the clinical center. Thirty-two subjects participated in the study and were randomly included in one of two treatment groups: 16 participants were included in the percutaneous electrolysis group (PE), and the remaining 16 patients were included in the TDN group (Figure 3), who were treated with dry needle puncture for four weeks in the epicondylar musculature. Patients in both groups were taught eccentric exercises for the epicondylar muscles. No differences between groups were observed at baseline for morphological characteristics or study variables (Table 1).

Significant differences were observed immediately post-treatment in the group treated with percutaneous electrolysis. Improvements were observed in both the patients’ pain perception (NPRS) and flexion movement, and the significant variables showed a high effect size (η2 between 0.199 and 0.466) (Table 2). The immediate clinical success post-treatment was 93.8% (15 patients) for the group treated with percutaneous electrolysis, and 75% (12 patients) for the TDN group.

The analysis also showed statistically significant differences in favor of the PE after one month of follow-up. A significant improvement was observed in the patients’ pain perception (NPRS), pressure algometry, and flexion movement. These variables showed a large effect size with η2 values between 0.118 and 0.465 (Table 3). The clinical success after one month of follow-up was 93.8% (15 patients) for the group treated with percutaneous electrolysis and 68.8% (11 patients) for the TDN group.

After three-months follow-up, statistically significant differences were still observed for the patients’ pain perception (NPRS), pressure algometry, and movement toward flexion of the elbow. Values of effect size (η2) were observed for these variables between 0.211 and 0.388 (Table 4). Considering a 25% improvement in the NPRS scale after three months of treatment as clinical success, this was 100% for the group treated with percutaneous electrolysis (16 patients) and 75% (12 patients) for the TDN group. If we consider a 50% improvement as clinical success, this was 93.8% for the group treated with percutaneous electrolysis and 43.8% for the TDN group.

## 4. Discussion

The main finding of the present study was that percutaneous electrolysis treatment added to eccentric exercises was more effective than trigger point dry needling added to eccentric exercises at improving pain (NPRS and PPT) and functionality (flexion range of motion) in patients with LE immediately post-treatment and at one and three-months of follow-up.

Despite having scarce evidence supporting it, PE is a popular technique among physicians in some countries. The first study that applied PE added to eccentric exercises to LE was by Valera-Garrido et al. [6]. This study found improvements of fifty points on the visual analog scale (VAS) after treatment that remained and were slightly improved at six weeks of follow up, results that are in line with those of the present study [6]. The main limitation of the study by Valera-Garrido et al. was the lack of a comparison group—a point solved in this study with the addition of a control TDN group [6]. The results obtained in the TDN group showed that both therapies were effective at improving pain in LE, but participants treated with PE improved more, obtaining a higher size effect and a clinical success immediately post-treatment of 93.8% versus 75% in the TDN group. The PPT at the LE also improved after treatment when compared to baseline; however, no significant differences were found between groups. Follow-up measures at one and three months demonstrate that effects remained in time, as Valera-Garrido et al. observed in their study [6]. Data yielded by this study also agreed with those regarding pain improvement obtained in other studies that used PE in other musculoskeletal disorders, such as subacromial pain syndrome [31], plantar fasciosis [32], and patellar tendinopathy [14]. Due to the previous results, percutaneous electrolysis could be considered an appropriate approach for diverse musculoskeletal pain syndromes.

Regarding function, PE improves elbow range of motion in flexion but not in extension. This could be explained by the fact that at baseline, all the patients reached the 0° extension range. Pronation and supination movements get increased after treatment, but not in a significant quantity, probably due to a complete range at baseline. Comparison with other studies cannot be performed as Valera-Garrido et al. did not measure function or range of motion; however, using ultrasound, they determined the reduction of degenerative structural changes observed in LE. Studies applying PE to other musculoskeletal pain conditions such as subacromial pain syndrome [31], plantar fasciosis [32], and patellar tendinopathy [14] showed significant improvements in function, which agree with those observed in this study. In terms of other secondary outcomes measured in this study, it must be noted that no improvements were observed in quality of life.

Despite being a minimally invasive technique, PE always has a higher risk than any other non-invasive technique. Several authors have investigated the technique assessing the risk of possible vasovagal reactions, which seems to be related with needling [33]. Nonetheless, the participants of the present study did not report any side effects.

The scarce evidence about PE in LE requires the development of futures studies comparing eccentric exercises added to PE versus eccentric exercises alone, which are the base treatment in all the studies. Due to the study designs of existing studies, it is not possible to differentiate how much improvement is due to each treatment. This study contributes to the existing evidence by comparison with another treatment technique, which is one step further from one arm designs; however, further studies are needed. Moreover, despite using a daily registry to monitor exercise compliance, it cannot be assured that participants correctly performed the proposed exercise daily. This aspect could influence the results due to the important role of this part of the treatment.

This study has several limitations. Firstly, a sham group would have been desirable to measure how much effect is due to participants’ expectations, although invasive sham therapy would have ethical implications. Moreover, the sample was limited, and multi-center studies with higher samples would provide more solid results. Regarding medication intake, participants were asked to avoid any change in their usual treatments in order to avoid possible side effects and/or alterations in their usual pain. Finally, a higher number of outcomes related to everyday activities would reveal how the improvements affected participants’ recoveries.

## 5. Conclusions

Ultrasound-guided percutaneous electrolysis as an adjunct to an eccentric exercise program is more effective for pain and range of movement than trigger point dry needling as an adjunct to the same exercise program in patients with lateral epicondylalgia.

## Figures and Tables

**Figure 1 jcm-09-02068-f001:**
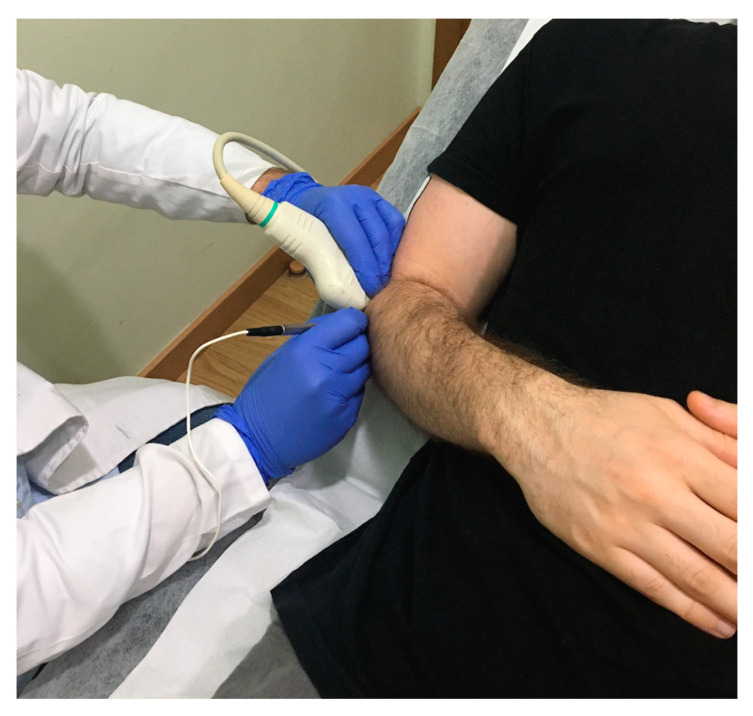
Application of percutaneous electrolysis (PE) in the lateral epicondyle.

**Figure 2 jcm-09-02068-f002:**
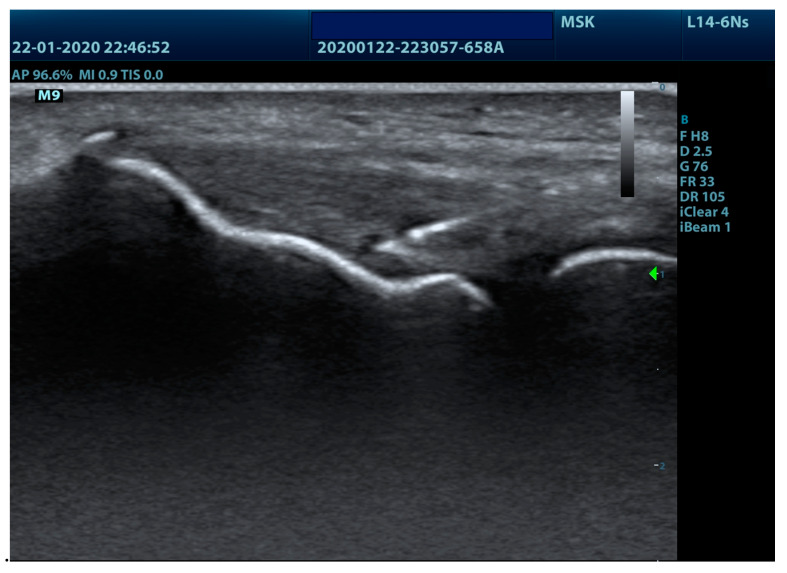
Ultrasound imaging of lateral epicondyle needling.

**Figure 3 jcm-09-02068-f003:**
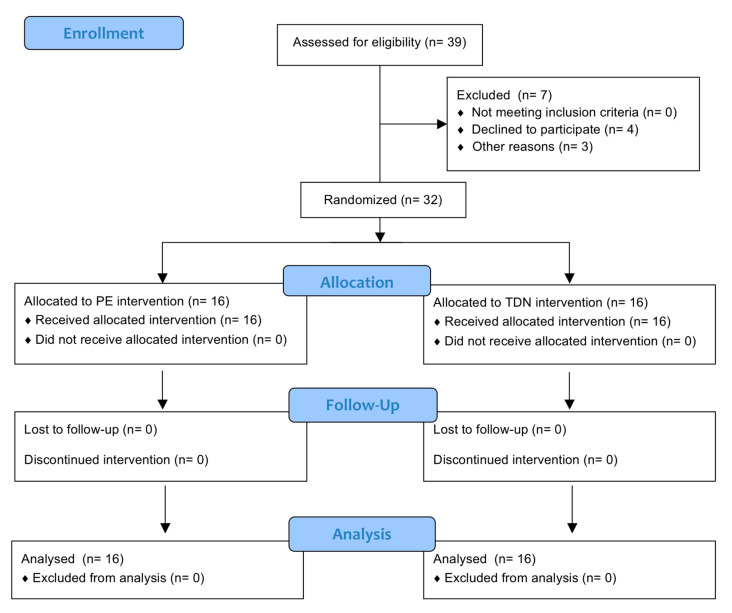
Flow diagram.

**Table 1 jcm-09-02068-t001:** Description of sample and groups at baseline.

Categorical		All (32)		PE Group (16)		TDN Group (16)		
		Frequency	%	Frequency	%	Frequency	%	
Gender	Male	20	62.5	10	62.5	10	62.5	
	Female	12	37.5	6	37.5	6	37.5	
Dominant hand	Right	29	9.4	15	93.8	14	87.5	
	Left	3	90.6	1	6.3	2	12.5	
Affected side	Right	27	84.4	14	87.5	13	18.8	
	Left	5	15.6	2	12.5	3	81.2	
**Continuous**		**Mean**	**SD**	**Mean**	**SD**	**Mean**	**SD**	***p***
Age (years)		38.16	13.89	40.44	15.51	35.88	12.12	0.361
Height (cm)		174.22	6.42	174.31	7.32	174.13	5.62	0.752
Weight (kg)		78.47	13.35	79.13	13.05	77.81	14.03	0.616
BMI		25.83	4.04	26.1	4.38	25.56	3.79	0.731
Flexion (°)		134.69	5.55	135.19	6.21	134.19	4.96	0.724
Extension (°)		−0.22	3.57	0.00	3.54	−0.44	3.71	0.633
Supination (°)		85.72	5.37	85.94	4.91	85.5	5.94	0.926
Pronation (°)		81.44	4.12	81.69	3.5	81.19	4.76	0.985
NPRS		5.91	1.59	6.44	1.21	5.38	1.78	0.058
PPT		3.38	0.53	3.36	0.53	3.39	0.56	0.809
PCS-SF12		48.68	5.97	48.83	6.18	48.52	5.94	0.867
MCS-SF12		54.56	3.99	55.16	3.78	53.97	4.24	0.41

PE: percutaneous electrolysis group; TDN: trigger point dry needling group; BMI: body mass index; MCS-SF12: Mental component of SF-12 questionnaire; NPRS: numerical pain rating scale; *p*: *p*-value; PCS-SF12: Physical component SF-12 questionnaire; PPT: pressure pain threshold; SD: standard deviation; °: degrees.

**Table 2 jcm-09-02068-t002:** Change scores, between-group differences in change score, and effect sizes of between-group differences in change scores at the end of treatment.

Continuous and Homoscedastic Variables								
		After Treatment		Within-Group Change Score		Between-Group Differences in Change Score		Effect Size
Variable	Group	Mean	SD	Mean	SD	F	*p*	η2
NPRS	PE	1.44	1.32	5.00	1.63	26.188	<0.001	0.466
	TDN	3.25	1.69	2.13	1.54			
**Non-Continuous and Homoscedastic Variables**								
		**After Treatment**		**Within-Group Change Score**		**Between-Group Differences in Change Score**		**Effect Size**
**Variable**	**Group**	**Mean**	**SD**	**Mean**	**SD**	**U**	***p***	**η2**
PPT	PE	4.33	0.60	0.96	0.67	78	0.057	0.111
	TDN	3.91	0.67	0.53	0.57			
MCS-SF12	PE	55.81	2.74	−0.65	2.83	106	0.404	0.021
	TDN	55.03	4.41	−1.06	2.08			
PCS-SF12	PE	51.09	6.13	−2.25	3.23	126	0.94	0.000
	TDN	50.34	4.59	−1.82	2.95			
Flexion	PE	139.25	4.80	4.06	3.75	61	0.006	0.199
	TDN	134.81	4.75	0.63	2.03			
Extension	PE	1.06	2.02	1.06	2.72	120	0.633	0.003
	TDN	0.19	3.73	0.63	1.71			
Supination	PE	88.31	3.20	2.38	4.52	125.5	0.911	0.000
	TDN	86.63	5.38	1.13	2.00			
Pronation	PE	83.75	2.24	2.06	3.07	99	0.167	0.037
	TDN	81.88	4.43	0.69	1.54			

MCS-SF12: Mental component of SF-12 questionnaire; NPRS: numerical pain rating scale; *p*: *p*-value; PCS-SF12: Physical component SF-12 questionnaire; PPT: pressure pain threshold; SD: standard deviation; U: Mann–Whitney U-test; η2: effect size eta-squared; TDN: trigger point dry needling group; PE: percutaneous electrolysis group.

**Table 3 jcm-09-02068-t003:** Change scores, between-group differences in change score, and effect sizes of between-group differences in change scores at one-month follow-up.

Continuous and Homoscedastic Variables								
		One-Month Follow-Up		Within-Group Change Score		Between-Groups Differences in Change Score		Effect Size
Variable	Group	Mean	SD	Mean	SD	F	*p*	η2
NPRS	PE	0.88	1.31	5.56	1.75	26.03	<0.001	0.465
	TDN	3.06	1.77	2.31	1.85			
PPT	PE	4.41	0.55	1.04	0.67	5.985	0.02	0.166
	TDN	3.90	0.68	0.51	0.55			
MCS-SF12	PE	55.52	3.20	−0.37	2.31	0.088	0.77	0.003
	TDN	54.59	4.38	−0.62	2.55			
**Non-Continuous and Homoscedastic Variables**								
		**One-Month Follow-Up**		**Within-Group Change Score**		**Between-Groups Differences in Change Score**		**Effect Size**
**Variable**	**Group**	**Mean**	**SD**	**Mean**	**SD**	**U**	***p***	**η2**
PCS-SF12	PE	52.75	5.07	−3.92	4.92	127	0.97	0.000
	TDN	52.05	3.42	−3.52	4.98			
Flexion	PE	140.13	5.06	4.94	5.17	76.5	0.04	0.118
	TDN	135.50	4.98	1.31	2.02			
Extension	PE	1.06	2.02	1.06	2.72	120	0.63	0.003
	TDN	0.19	3.73	0.63	1.71			
Supination	PE	88.94	2.72	3.00	4.80	110	0.42	0.014
	TDN	86.63	5.38	1.13	2.00			
Pronation	PE	83.75	2.24	2.06	3.07	104	0.27	0.026
	TDN	81.94	4.40	0.75	1.53			

MCS-SF12: Mental component of SF-12 questionnaire; NPRS: numerical pain rating scale; *p*: *p*-value; PCS-SF12: Physical component SF-12 questionnaire; PPT: pressure pain threshold; SD: standard deviation; U: Mann–Whitney U-test; η2: effect size eta-squared; TDN: trigger point dry needling group; PE: percutaneous electrolysis group.

**Table 4 jcm-09-02068-t004:** Change scores, between-group differences in change score, and effect sizes of between-group differences in change scores at 3-months follow-up.

Continuous and Homoscedastic Variables								
		Three-Month Follow-Up		Within-Group Change Score		Between-Group Differences in Change Score		Effect Size
Variable	Group	Mean	SD	Mean	SD	F	*p*	η2
MCS-SF12	PE	56.17	2.449	−1.017	2.348	0.41	0.528	0.013
	TDN	54.52	4.378	−0.545	1.794			
**Non-Continuous and Homoscedastic Variables**								
		**Three-Month Follow-Up**		**Within-Group Change Score**		**Between-Group Differences in Change Score**		**Effect Size**
**Variable**	**Group**	**Mean**	**SD**	**Mean**	**SD**	**U**	***p***	**η2**
NPRS	PE	0.38	0.89	6.06	1.44	34.5	<0.001	0.388
	TDN	2.56	2.19	2.81	2.37			
PPT	PE	4.41	0.55	1.15	0.66	53.5	0.004	0.246
	TDN	3.90	0.68	0.51	0.56			
PCS-SF12	PE	53.69	3.34	−4.86	5.54	116	0.637	0.008
	TDN	52.20	3.49	−3.68	4.92			
Flexion	PE	139.06	4.55	3.88	4.19	59	0.003	0.211
	TDN	134.13	3.49	−0.63	1.91			
Extension	PE	1.06	2.02	1.06	2.72	118	0.584	0.005
	TDN	2.06	3.07	2.50	5.48			
Supination	PE	88.75	2.89	2.81	4.90	118	0.63	0.005
	TDN	86.56	5.39	1.06	2.02			
Pronation	PE	83.75	2.24	2.06	3.07	88.5	0.06	0.069
	TDN	81.56	4.37	0.38	1.54			

MCS-SF12: Mental component of SF-12 questionnaire; NPRS: numerical pain rating scale; *p*: *p*-value; PCS-SF12: Physical component SF-12 questionnaire; PPT: pressure pain threshold; SD: standard deviation; U: Mann–Whitney U-test; η2: effect size eta-squared; TDN: trigger point dry needling group; PE: percutaneous electrolysis group.

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
