# Peer review of "Percutaneous Electrolysis in the Treatment of Lateral Epicondylalgia: A Single-Blind Randomized Controlled Trial"

_jcm, 2020, doi:10.3390/jcm9072068_

Round 1

Reviewer 1 Report

I congratulate the authors on a well conducted trial. However, I suggest that a revision of the English language writing style be conducted to give more clarity to the manuscript. My other suggestions are below.

Abstract:are the results stated a within group or between group analysis? The last sentence should clarify, in combination with an eccentric exercise routine as per the first sentence in the discussion section.

Introduction:

Line 37: should this be “it is most prevalent among individuals aged 35-50”?

Line 51: change to …”and should include specific tests such as…? (7).” List some of the tests.

Line 63 Change LA to LE

Methods

Line 90: change to “conventional physiotherapy consisting of…” You need to explain how this is different to the eccentric exercise routine.

Sample size calculation: consider including the actual primary outcome measure used to calculate this and magnitude of the difference between groups that you are looking to detect.

Line 158-165: did you collect compliance to the eccentric exercise routine? This is important and should be reported.

Line 186-190: how was this percentage change calculated? Was there any adjustment to account for where on the scale the baseline score lay? Please provide more detail.

Statistical analysis: was there any consideration for baseline scores as covariates in your analysis? The NPRS scores are very close to being statistically different.

Results

Consider reporting compliance to the exercise regime.

Discussion

How could exercise compliance influence the results? If you don’t have this data then this should be added to the limitations section.

Conclusion

Consider change to; "Ultrasound-guided percutaneous electrolysis as an adjunct to an eccentric exercise program is more effective for pain and range of movement than trigger point dry needling as an adjunct to the same eccentric exercise program."

Reviewer 2 Report

The article includes the results of a randomised controlled trial having in view the treatment of patients with lateral epicondyalgia.

Abstract. Refer if NPRS and VAS involve the same assessment.

Introduction. Explain if the degenerative mechanism is the only one involved in the appearance of lateral epicondyalgia. What specific tests are used in oder to perform the physical examination in patients with lateral epicondyalgia? The statement "...producing its stimulation by an accute inflammatory response" should be more accurately explained.Explain the abbreviation LA (line 63).

Material and methods. The abbreviation should be explained first time they appear in the manuscript (NPRS, line 90). The outcome measures PCS-SF12 and MCS-SF12 are not described.

Results. Table 1: include also the measurements units.

Discussion. Are there studies that compared the efficiency of exercise programme versus percutaneous electrolysis plus exercise? Refer also to this aspect.

Conclusions should be temperated as the study included only 32 patients.

Round 2

Reviewer 2 Report

The authors answered all my comments.